# Large-scale 2D heterostructures from hydrogen-bonded organic frameworks and graphene with distinct Dirac and flat bands

Xin Zhang [1,6], Xiaoyin Li [2,6], Zhengwang Cheng [3,6], Aixi Chen[4,6], Pengdong Wang[4,6], Xingyue Wang[1], Xiaoxu Lei[4], Qi Bian[5], Shaojian Li[5], Bingkai Yuan[4], Jianzhi Gao [1] ✉, Fang-Sen Li [4] ✉, Minghu Pan [1,5] ✉ & Feng Liu [2] ✉

The current strategies for building 2D organic-inorganic heterojunctions involve mostly wet-chemistry processes or exfoliation and transfer, leading to interface contaminations, poor crystallizing, or limited size. Here we show a bottom-up procedure to fabricate 2D large-scale heterostructure with clean interface and highly-crystalline sheets. As a prototypical example, a well-ordered hydrogen-bonded organic framework is self-assembled on the highly-oriented-pyrolytic-graphite substrate. The organic framework adopts a honeycomb lattice with faulted/unfaulted halves in a unit cell, resemble to molecular "graphene". Interestingly, the topmost layer of substrate is self-lifted by organic framework via strong interlayer coupling, to form effectively a floating organic framework/graphene heterostructure. The individual layer of heterostructure inherits its intrinsic property, exhibiting distinct Dirac bands of graphene and narrow bands of organic framework. Our results demonstrate a promising approach to fabricate 2D organic-inorganic heterostructure with large-scale uniformity and highly-crystalline via the self-lifting effect, which is generally applicable to most of van der Waals materials.

Heterostructures have been widely used as a basic building block in the advanced semiconductor devices owing to their essential and attractive structural, interfacial, and electronic properties, especially for a comparatively better electrical and optical performance with respect to individual materials. In recent years, two-dimensional (2D) heterojunctions are vastly developed soon after the rise of graphene[1] and other 2D materials[2–7]. The organic−inorganic heterostructures may take the advantages of both organic and inorganic materials, for example to make it flexible like an organic film and at the same time wearable like an inorganic film. The combination of their different properties and functionalities may broaden the device's capabilities and endow special applications that neither one can achieve alone. By combining the organic layers with a dangling-bond-free surface of inorganic van der Waals (vdW) materials[8–11], 2D organic-inorganic heterojunctions have emerged and become new paradigms with presenting unprecedented multifunctional application[12–16] and novel device architectures[17–20]. Currently, the strategies for building 2D organic-inorganic heterojunctions include, i.e., solution processing[21,22], mechanical exfoliation and transfer[23–26], and vapor phase growth[27]. The first method involves inevitably the contamination from wet-chemistry while the second has limited the size of heterojunction (~micrometers). In contrast, organic vapor phase growth can grow a highly-ordered

[1]School of Physics and Information Technology, Shaanxi Normal University, Xi'an 710119, China. [2]Department of Materials Science and Engineering, University of Utah, Salt Lake City, UT 84112, USA. [3]School of Science, Hubei University of Technology, Wuhan 430068, China. [4]Vacuum Interconnected Nanotech Workstation, Suzhou Institute of Nano-Tech and Nano-Bionics, Chinese Academy of Sciences (CAS), Suzhou 215123, China. [5]School of Physics, Huazhong University of Science and Technology, Wuhan 430074, China. [6]These authors contributed equally: Xin Zhang, Xiaoyin Li, Zhengwang Cheng, Aixi Chen, Pengdong Wang. ✉e-mail: jianzhigao@snnu.edu.cn; fsli2015@sinano.ac.cn; minghupan@snnu.edu.cn; fliu@eng.utah.edu

organic layer on the substrate surface and produce high-quality heterostructures[28], such as the organic layers on the surface of graphene[29,30], WS$_2$[31], and MoSe$_2$[32]. However, the residuals at the interface from vapor phase growth strongly affect the quality of heterostructures[33,34]. Up to now, a rational bottom-up design of 2D high-quality organic-inorganic heterojunctions with the appealing properties of both organic and inorganic layers is rarely demonstrated.

Here, we develop an approach to bottom-up fabricate 2D large-scale organic-inorganic heterostructure with cleanest interface and highly-crystalline layer structure, all carried out in ultrahigh vacuum (UHV) environment. The heterostructure is composed of a monolayer 1,3,5-tris(4-hydroxyphenyl)benzene (THPB)-hydrogen-bonded organic framework (HOF) and the graphene layer made by self-lifting the topmost layer from highly-oriented-pyrolytic-graphite (HOPG) via strong interlayer coupling. By utilizing in-situ high resolution scanning tunneling microscopy/spectroscopy (STM/STS) and angle-resolved photoelectron spectroscopy (ARPES), we observed a honeycomb THPB-HOF lattice with faulted/unfaulted halves, Dirac bands near $E_F$ and a series of narrow bands at deeper energies. Dirac bands observed in the energy range of >1.0 eV below $E_F$ with high fermion velocity $\approx$4.83–5.25 eV·Å, can be attributed to the self-lifted graphene layer. In addition, narrow bands are originated from THPB-HOF lattice, as a unique form of molecular "graphene"[35], in agreement with DFT calculated band structure consisting of topological flat bands. We also observed local spin states located slightly above the $E_F$ in the tunneling spectra, induced by removing $p_z$ orbitals in $\pi$-conjugated THPB systems. Our results demonstrate the feasibility of fabricating large-scale 2D organic framework/graphene heterostructure, with large-scale uniformity and long-range order. In short, we present a simple effective method to make freestanding 2D large-scale organic-inorganic heterostructure by a self-lifting effect, as confirmed by directly observing the coexistence of distinct Dirac bands with high carrier mobility from graphene and flattened bands of HOF due to enhanced organic-inorganic interaction. We also further demonstrated the generality of our method by growth on MOS$_2$ substrate, where we observed the bandgap transition of MoS$_2$ from bulk to monolayer via again the self-lifting effect induced by THPB-HOF monolayer.

## Results

### Self-lifting a large-scale graphene from graphite by molecular overlayer

A simple but effective approach for fabricating the large-scale 2D organic/graphene heterostructure is illustrated in Fig. 1a. First, the exfoliation of HOPG provides a clean, atomic flat surface; Second, the deposition of molecules, following with in-situ annealing, allows for the self-assembly of THPB molecules via hydrogen-bonded; At last, the formation of HOF monolayer provides extra force to lift the topmost graphite layer (graphene) from the substrate. All steps are conducted in UHV, giving a contamination-free interface and other notable advantages, i.e. large-scale uniformity, and high quality of both organic and graphene layers, as demonstrated below. The key procedure is the self-lifting of topmost graphite by a moderately-strong interlayer interaction between organic layer and HOPG.

To demonstrate the feasibility of this approach, we conducted both experiments and DFT calculations. A THPB-HOF monolayer is grown on HOPG substrate (see Methods for sample preparation), and STM images (Fig. 1b and Supplementary Fig. 4) display large-scale uniformity and long-range order of THPB-HOF layer. Height profile measurements in Fig. 1b, c show the distances of HOF/topmost (1st) layer and HOF/1st/second (2nd) layer of HOPG are about 2.65 and 6.48 Å, respectively, giving the distance between the 1st and 2nd layer of HOPG to be about 3.83 Å, significantly larger than the interlayer separation of HOPG (3.34 Å). This is also confirmed by the DFT calculation (Fig. 1d, e and Supplementary Fig. 14). Here the four-layered HOPG is used as a manifestation, but we emphasize that the self-lifting

effect of THPB-HOF monolayer and the resultant exfoliation of graphene are robust regardless of the thickness of HOPG (see Supplementary Fig. 14). The optimized structure composed of THPB-HOF and four graphite layers, shows a reduced distance $\approx$2.71 Å between HOF and the 1st layer, and the enlarged distance $\approx$3.41 Å between the 1st layer and 2nd layer of HOPG, indicating a stronger coupling between HOF and 1st layer, as well as an effective self-lifting of graphene layer from HOPG substrate. The calculated band structure of the resultant HOF/graphene heterostructure (Fig. 1f) shows the coexistence of both Dirac band of graphene band-folded into the Brillouin zone (BZ) of HOF and narrow bands of THPB-HOF, which serve as the electronic evidence for validating the formation of large-scale floating 2D THPB-HOF/graphene heterostructure, in agreement with the ARPES measurements. We also tried deposit THPB molecule on in-situ-cleaved MoS$_2$ surface. Supplementary Figs. 11 and 12 show the morphology of THPB molecules on MoS$_2$ surface measured by STM. The thickness of monolayer THPB framework is measured about 2.94 Å, similar to 2.65–2.71 Å on HOPG substrate. High resolution STM image in Supplementary Fig. 12b shows the similar HOF structure with unit cell size about 2.67 nm, larger than the size of unit cell of THPB-framework on HOPG (16.5 Å). Most notably, we measured differential conductance spectroscopy (d$I$/d$V$) on both clean MoS$_2$ surface and THPB molecule island (Supplementary Fig. 12c, d). The bandgap of MoS$_2$ clean surface is measured about 1.46 eV, slightly smaller than the bandgap (1.78 eV) measured on THPB-framework, indicating the self-lifting effect of THPB-framework acts also on the MoS$_2$ layer. The bulk MoS$_2$ material is reported to have an indirect bandgap of 1.2 eV, whereas two-dimensional (2D) single-layer MoS$_2$ nanosheets have a direct bandgap of 1.8 eV[36,37]. Our results are consistent with the reported bandgap transition of MoS$_2$ from bulk to monolayer. These results evidently support the generality of our method.

### Growth and characterization of large-scale uniform, highly-ordered 2D THPB-HOF on HOPG

The model in Fig. 2a gives the resulted THPB-HOF structure with the non-centrosymmetric space group P6 (No. 168). Previous studies showed that THPB molecules can form self-assembled superstructure on Au(111) and Ag(111) surfaces[38–40]. High resolution image (Fig. 2c) reveals a honeycomb lattice with two edge sharing THPB molecules *per* unit cell, distinct from the corner-sharing triangular lattice of THPB on Au(111)[40]. In Fig. 2b, d, each Shuriken-shaped feature is a single THPB molecule, in which three peripheral benzene rings show bright contrast while the central benzene is invisible in STM images. The images of a triangular defect (Fig. 2b, d) provide a hint for the proposed structure, where the size and shape of the black triangular defect resemble a missing THPB molecule.

The optimized structure of THPB-HOF/graphene from ab initio DFT calculations is shown in Fig. 2e and Supplementary Fig. 2, which reveals that the constituting molecules with a planar conformation are linked to each other via O…H-O hydrogen bond between hydroxyl groups at the corners. The calculated unit-cell lattice constant is 17.23 Å, consistent with the experimental value of 16.5 Å. The distance between the THPB-HOF and the 1st carbon layer is 2.65 Å (Fig. 1c), indicating a moderately stronger interlayer coupling than vdW interaction. In the unit cell, four phenyl rings of a THPB sit on the top of carbon rings in the underneath graphene layer (Bernal stacking), named "unfaulted" half; while phenyl rings of another THPB are centered around a C atom of the underneath graphene layer (A-B or A-C stacking), named "faulted" half (Fig. 2e). By overlaying the relaxed model on top of the STM image (Fig. 2f), bright protrusions in the STM image correspond to the tilted corner phenyl rings of THPB with the resolved intermolecular structure. Similar contrast of THPB-HOF can be clearly visualized in STM images with different biases in Supplementary Fig. 5. More interestingly, we also observed two different kinds of chirality for H-bond hollow-ring in STM imaging of THPB-HOF,

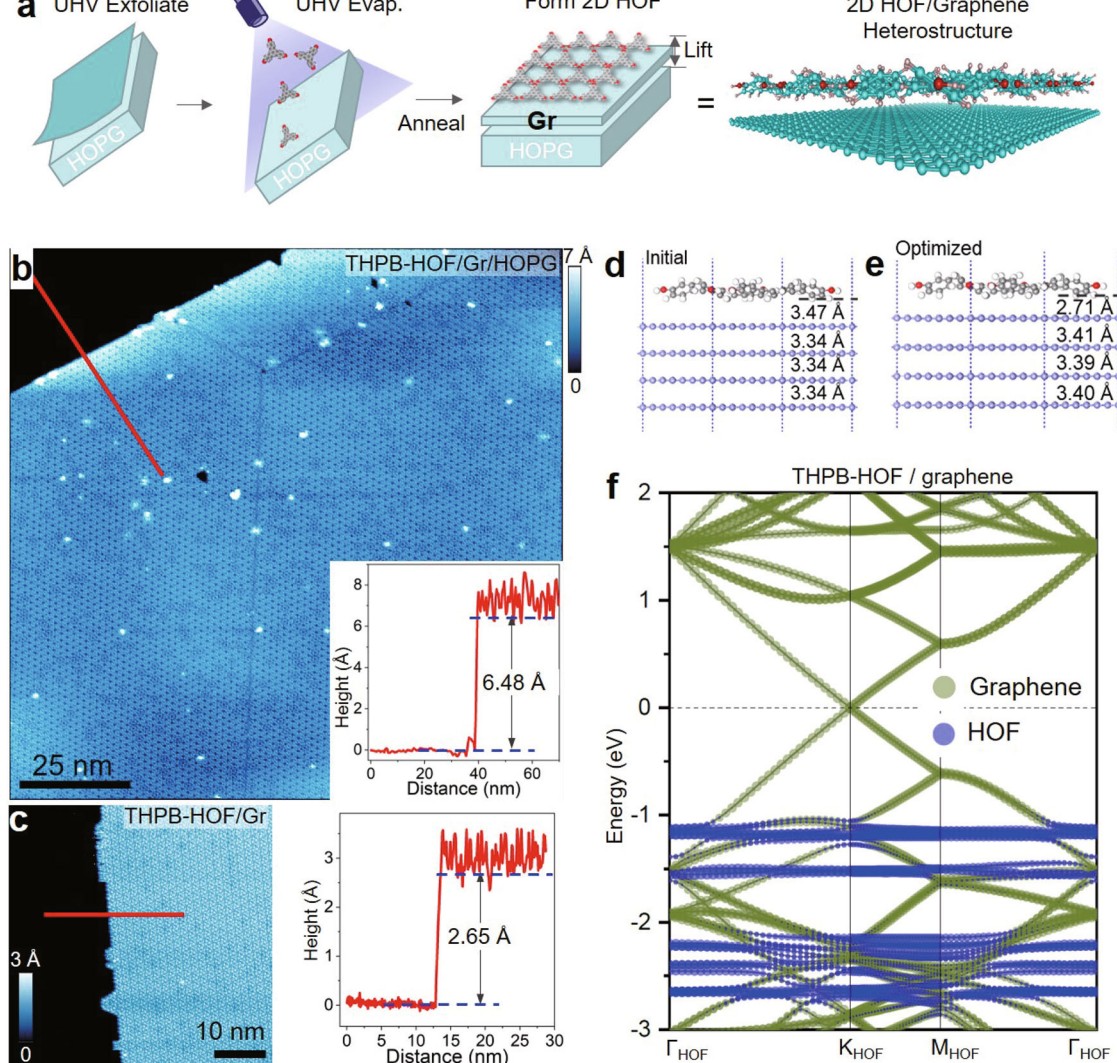

**Fig. 1 | Bottom-up fabrication of large-scale 2D organic/graphene hetero-structure. a** Schematic drawing to show three-step processes for the formation of 2D HOF and the lifting of topmost graphene layer from HOPG substrate. STM measurements of a large-scale uniform and highly ordered THPB-HOF monolayer and the height profiles measured for THPB-HOF/Gr/HOPG (**b**) and THPB-HOF/Gr (**c**), respectively. The image sizes in **b** and **c** are $125 \times 125$ nm$^2$ and $50 \times 50$ nm$^2$ with the setting parameters of $V_B = -2.0$ V and $I_T = 10$ pA. The slab calculation of the HOF monolayer on top of four graphite layers, showing the initial (**d**) and final optimized (**e**) structures. The color of atoms: dark gray (C in THPB), white (H), red (O) and blue (C in graphite), respectively. **f** The calculated band structure of the optimized HOF/graphene heterostructure within the BZ of THPB-HOF, in which the Dirac band from the lifted graphene layer (green) and narrow bands from THPB-HOF (blue) are seen to coexist.

as shown in Supplementary Fig. 6a, b. Interestingly, by viewing each THPB as a "superatom", the THPB-HOF structure has basically a graphene-like structure with hydrogen bonds connecting the THPBs, instead of covalent C-C bonds in graphene (left-lower panel of Fig. 2a). Two sublattices are attributed to THPB of the faulted and unfaulted halves, which breaks in principle the inversion symmetry and gives rises to an on-site energy difference. The in-situ Raman spectrum of THPB-HOF/HOPG displays the vibrational modes of H-bonds (Supplementary Fig. 15).

**ARPES observation of Dirac and narrow bands in THPB-HOF/HOPG**

The mesoscale ordered 2D THPB-HOF has enabled the ARPES characterization of its band structure. As a reference, the bare HOPG surface was measured firstly (Supplementary Note 3 and Fig. 7a). In the ranges of 0 ~ −2.0 eV and ±0.6 Å$^{-1}$, the bare HOPG surface has no band, to provide a clean background for observing the THPB-HOF bands. Figure 3c shows high-resolution ARPES of nearly-full coverage THPB-

HOF/HOPG measured at 77 K. Multiple linear dispersive bands can be clearly seen in the energy range from −1.2 eV to E$_F$, which are better resolved in the second-derivative intensity plot (Fig. 3d), marked as $\lambda_1$ to $\lambda_6$, not presented in ARPES of bare HOPG. The slopes (Fermi velocity) of bands $\lambda_1$ to $\lambda_6$, are estimated to be ≈4.83–5.25 eVÅ ($\hbar v_F \approx 7.33$–$7.96 \times 10^5$ ms$^{-1}$), comparable with that of graphene ($1.0 \times 10^6$ ms$^{-1}$)[41]. Each linear band expands about 0.25 Å$^{-1}$ in momentum space. Note that the calculated lattice constant of THPB-HOF (17.23 Å) is about seven times of that of graphite (2.46 Å) and the lattice vectors $a_1$ and $a_2$ are aligned parallel with the underlying graphite lattice (Fig. 3a). Considering the size of the graphite BZ (1.7 Å$^{-1}$ from Γ to K), the deduced BZ of THPB-HOF is about 0.243 Å$^{-1}$. To explore the origin of these linear bands, the band structure of THPB-HOF/graphene was calculated in a range of $-0.2 - 1.05$ Å$^{-1}$ along the $\Gamma_{HOF}$-$K_{HOF}$ direction. The calculated result (Fig. 3e and Supplementary Fig. 3) shows multiple linear bands right below the Fermi level, matching nicely with the dispersion of observed linear bands. According to the component analysis of bands as shown in Fig. 1e, here the calculated

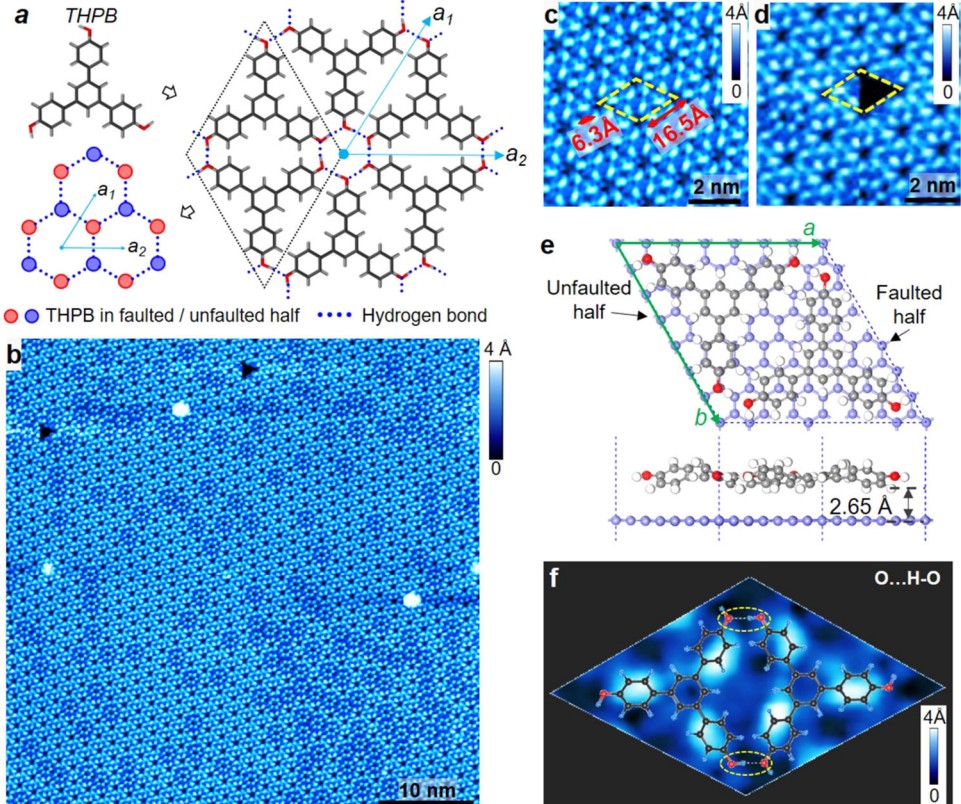

**Fig. 2 | STM characterization and ab initio DFT calculation of THPB-HOF. a** The stick models of single THPB molecule and THPB-HOF superstructure. THPB molecules are assembled via O...H-O hydrogen bonding between hydroxyl groups at the corners. Blue dotted lines show the hydrogen bonds. The dotted rhombus indicates the unit cell of THPB-HOF. **b** Topographic STM image show the large-scale self-assembled THPB-HOF on HOPG substrate. The image size is $50 \times 50\,nm^2$ with $V_B = -1.5\,V$ and $I_T = 10\,pA$. **c** Zoom-in image shows the size of unit cell is about 16.5 Å. The image is $10 \times 10\,nm^2$ with $V_B = -1.0\,V$ and $I_T = 10\,pA$. **d** High resolution image showing a single THPB vacancy. **e** Top (upper panel) and side (lower panel) view of the optimized structure of THPB-HOF on top of graphene. The white, red, gray and blue balls represent the hydrogen, oxygen, carbon atoms in THPB molecules and the carbon atoms in graphene, respectively. The dashed line indicates the unit cell. **f** High resolution STM image displaying the unit cell of THPB-HOF, overlaid with the ball-and-stick model of THPB molecules. Yellow dashed ovals indicate the hydrogen bond of O...H-O.

linear bands are originated from the graphene layer of the THPB-HOF/ graphene heterostructure. Instead, two $\pi$ bands of HOPG, the upper ($\pi_1$) and the lower $\pi$ band ($\pi_2$) with the splitting at the K point $\approx 0.5\,eV$, are distinct from the observed linear bands, and hence can be ruled out. Furthermore, the constant-energy contour (CEC) at $-0.1\,eV$ in Fig. 3b, signifies the presence of multiple linear bands. A hexagonal ring (band $\lambda_1$) is clearly visible around the zone center and a series of bands ($\lambda_2$ to $\lambda_6$) extend from the center to the zone boundary with equally spacing about $0.24\,Å^{-1}$. Based on these observations, the bands $\lambda_1$ to $\lambda_6$ can be assigned to the Dirac bands of topmost graphene, folded into the BZ of THPB-HOF.

In addition, at deeper energies, various non-dispersive bands emerge in ARPES intensity (Fig. 3g) and the second-derivative (Fig. 3h) plot, namely bands $\delta_1$ to $\delta_5$ located at the energies of $-2.2$, $-2.8$, $-3.4$, $-4.0$ and $-5.3\,eV$, respectively. Note that these narrow bands are not appearing in ARPES of bare HOPG surface (Supplementary Fig. 7a), so that they are attributed to THPB-HOF monolayer. Our previous work shows topological flat band[40] in 2D HOF self-assembled on Au(111) substrate with large-scale uniformity and long-range order. By employing DFT-based calculations to calculate the band structure of the freestanding THPB-HOF monolayer (Fig. 3f), we find a number of narrow bands appearing in both conduction and valence states, in consistence with our ARPES observation. Among these bands, two topmost sets of valence bands can be identified as two kinds of topological bands with small bandwidth (Fig. 3f, i). One is dual topological flat bands[42,43] in THPB-HOF hexagonal lattice. Another is Dirac

bands resembling to the case of molecular "graphene", indicating that the THPB-HOF is effectively a $(p_x, p_y)$-graphene[44]. The ARPES data and the calculated band structure of THPB-HOF are in a good agreement, despite the binding energy difference of these narrow bands, which is probably caused by the existence of long-range Coulomb in this system[45]. We also reproduce THPB-HOF on the surface of bilayer graphene (BLG) on SiC substrate. Similar Dirac bands from BLG and narrow bands from THPB-HOF can be observed in ARPES (Supplementary Fig. 16).

## DFT calculations of topological flat bands and Dirac bands of THPB-HOF

Topological Dirac bands[46–50] and flat bands[51–53] have been predicted with specific lattice symmetries[54–57], in 2D covalent organic frameworks (COFs)[57–59] and metal–organic frameworks (MOFs)[60–66]. To explore the intrinsic electronic property of THPB-HOF, we performed DFT calculations to obtain its band structure in the freestanding form (see Methods). The geometric structure of THPB-HOF monolayer for DFT calculations is shown in Supplementary Fig. 14, where the experimental lattice constants ($a = b = 16.5\,Å$) are adopted. The calculated band structure in Fig. 3f shows that THPB-HOF is a semiconductor with a large bandgap of $3.28\,eV$ (by local density approximation method) and a series of nearly-flat bands with very narrow dispersion below the Fermi level ($E_F$). Considering the charge transferring between HOPG substrate and THPB-HOF, the $E_F$ will locate around the midpoint of the bandgap; as a result, the energy positions of these calculated

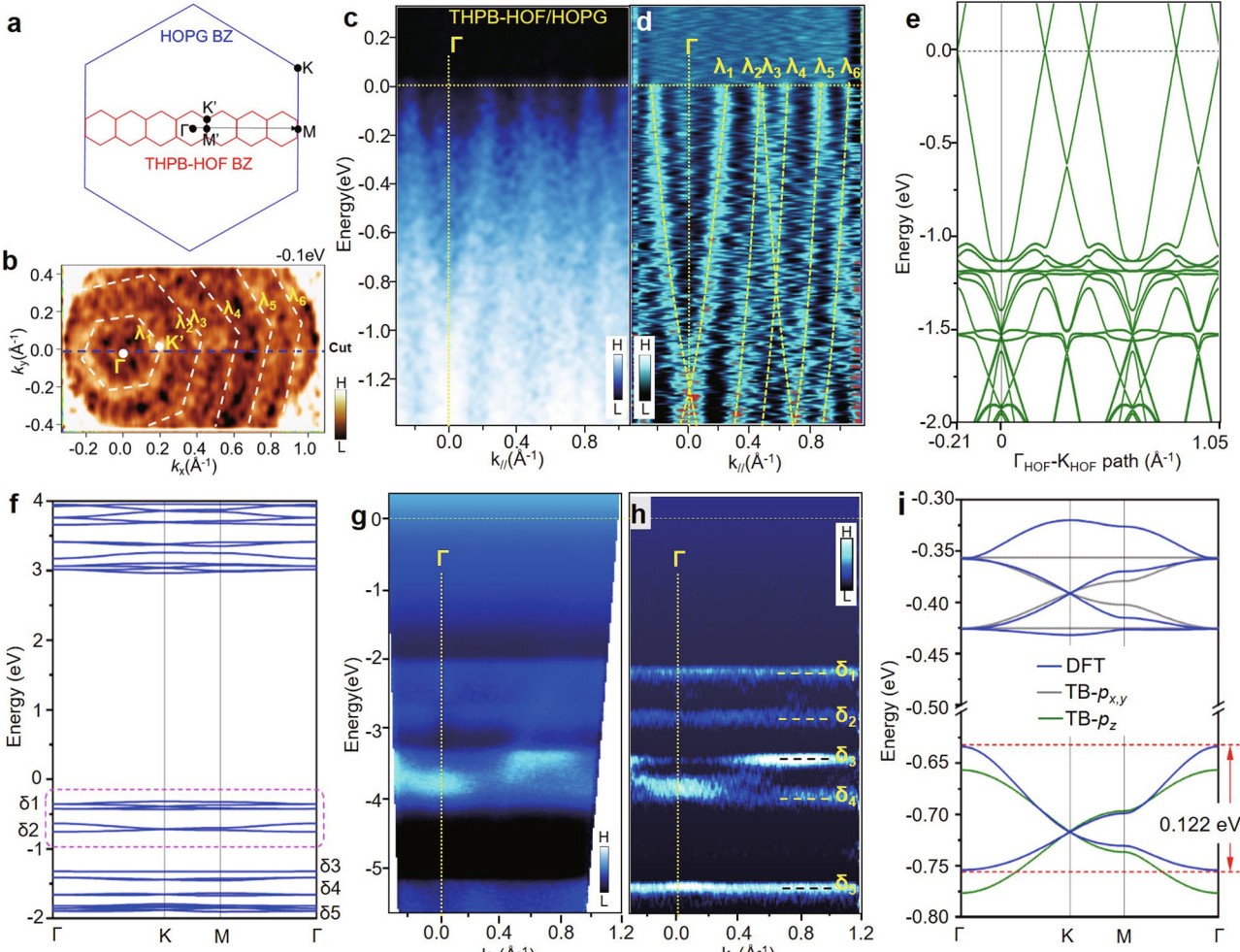

**Fig. 3 | ARPES observation of the THPB-HOF/graphene bands. a** The relationship between the BZs of HOPG surface and THPB-HOF. Note, the lattice constant of THPB-HOF (17.0 Å) is about seven times of that of graphite (2.46 Å) and the lattice vectors $a_1$ and $a_2$ are aligned parallel with the graphite lattice. **b** The CEC measured at −0.1 eV and 77 K. The hexagonal-shaped linear bands, $\lambda_1$ to $\lambda_6$ of THPB-HOF are clearly visible. The blue dashed lines indicate the line-cut taken for measurement in **c** and **d**. High-resolution ARPES spectra (**c**) and second-derivative intensity plot (**d**), taken along $\Gamma$-$K_{//}$ direction from a nearly-full coverage of THPB-HOF. A series of linear dispersive bands can be clearly resolved, denoted as $\lambda_1$ to $\lambda_6$ in the momentum. **e** The calculated band structure of THPB-HOF/graphene along $\Gamma$-$K$ direction from −0.21 to 1.05 Å$^{-1}$. **f** The calculated band structure within the energy window from −2.0 to +4.0 eV. The optimized structure of THPB-HOF using the

experimental lattice constants of a = b = 16.5 Å. The topmost two sets of valence bands are highlighted by the purple dashed rectangle, and a zoomed-in view is presented in **i**. The upper bands are dual topological flat bands in hexagonal lattice, while the lower bands are Dirac bands resembling the case of graphene. The blue, gray and olive bands are DFT, $p_{x,y}$-orbital and $p_z$-orbital TB model results, respectively. The fitting NN hopping strengths for the $p_{x,y}$-orbital and $p_z$-orbital TB models are 0.023 and 0.020 eV. The large energy scaled ARPES spectra (**g**) and second-derivative intensity plot (**h**), taken from −5.5 eV to 0 eV for a full-coverage THPB-HOF film. A series of non-dispersive bands can be clearly resolved at the energies below −2.1 eV. "H" and "L" in color scale is for high/low intensity of ARPES signal, respectively.

THPB-HOF bands will appear far below $E_F$, corresponding to the bands $\delta_1$ to $\delta_5$ in ARPES. One salient feature of these THPB-HOF bands is the narrow bandwidth, especially for valence bands near the Fermi level, which is consistent with nearly non-dispersive bands experimentally observed in the ARPES. Surprisingly, we find that the topmost valence bands are exactly two kinds of topological bands (Fig. 3i), namely, the upper dual topological flat bands and the lower Dirac bands. The former and the latter can be captured by the $p_{x,y}$-orbital and $p_z$-orbital tight-binding (TB) models in the hexagonal lattice respectively. By just considering the nearest-neighbor (NN) hopping, both TB models can well reproduce the DFT results as shown in Fig. 3i, validating the two sets of topological bands. Here, the velocity of Dirac bands is estimated to be 0.252 eV·Å (see Supplementary Fig. 8 for details), much smaller than that of graphene ($\approx$6.6 eV·Å)[67]. The reduced carrier velocity and narrow bandwidth (0.122 eV) are attributed to the large cell size and weak hopping via hydrogen bonds of THPB-HOF. Narrow electronic

energy bands provide an opportunity to explore many-body quantum phases of matter, resulting in a wealth of correlated, topological and broken-symmetry phases. For examples, the upper dual topological flat bands can give rise to exotic collective behaviors in the presence of Coulomb interactions when the bands are partially filled (Supplementary Note 8 and Fig. 10), like the emergence of magnetism and superconductivity in twisted bilayer systems[68–70].

## Tunneling spectra measured on THPB-HOF
The d$I$/d$V$ spectroscopic measurements, proportional to local density of states, were performed on THPB-HOF at 77 K (Fig. 4a). As shown in Fig. 4b, the d$I$/d$V$ spectra show a typical V-shaped density-of-state (DOS) with the neutrality point near $E_F$ at the locations of hollow sites, as blue-shaded area in Fig. 4c. When the tip approaching on the top of THPB molecule, a prominent DOS peak appears at the energy of +27 meV with the half-peak width of 60 meV in d$I$/d$V$ spectra.

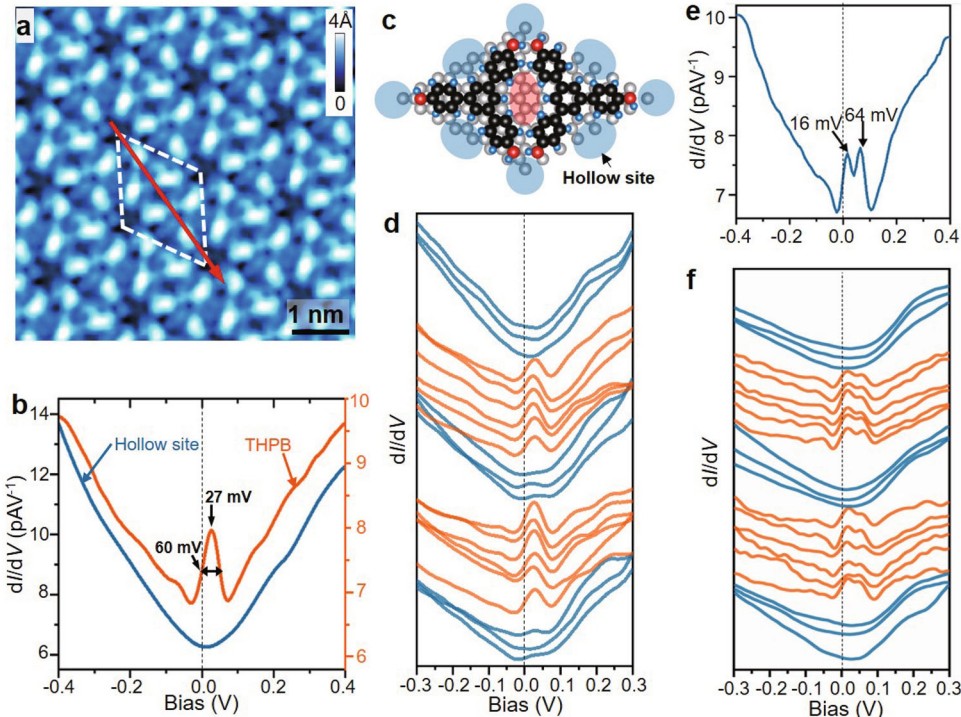

**Fig. 4 | Tunneling spectra measured on THPB-HOF. a** Topographic image of THPB self-assembled structure. The image size is $5.5 \times 5.5\ nm^2$ with $V_b = -1.0\ V$ and $I_t = 20\ pA$. White dashed diamond indicates a unit cell of THPB-HOF. Red arrow shows the trace of d$I$/d$V$ spectroscopic survey. **b** Two representative d$I$/d$V$ spectrum taken on the hollow sites (blue curve) and on the THPB molecule (red curve), respectively. **c** The ball-and-stick model shows the hollow sites in the top view. **d** A d$I$/d$V$ spectroscopic line survey. **e** Another set of d$I$/d$V$ spectrum taken on the THPB molecule with showing a double-peak feature located at +16 and +64 mV, respectively. **f** A d$I$/d$V$ spectroscopic line survey with showing the double peaks evolving within the unit cell. All d$I$/d$V$ spectra were measured with $V_B = 0.3\ V$, $I_T = 200\ pA$ and a bias modulation of 7 mV at the temperature of 77 K.

A delicate, line d$I$/d$V$ spectroscopic survey (Fig. 4d) measured within a unit cell of THPB-HOF along red line marked in Fig. 4a, shows that the emerging DOS peak is highly correlated with THPB-HOF. More interestingly, a double-peak feature appears occasionally, as shown in Fig. 4e, in which two DOS peaks lie at the energies of +16 and +64 meV, respectively, with the energy separation of about 48 meV. The line STS survey (Fig. 4f) also indicates that such double-peak feature is also observed exclusively on THPB molecules.

Such DOS peaks near $E_F$, has been observed previously in the chemisorption of single hydrogen[71] or boron[72] on graphene, indicating a local magnetic state introduced by removing $p_z$ orbitals in $\pi$-conjugated carbon systems. Despite pristine graphene is intrinsically nonmagnetic due to its $\pi$-conjugated electrons, Lieb[73] proposed that the ground state of graphene may possess a total spin given by S = $1/2 \times |N_A - N_B|$, where $N_A$ and $N_B$ are the number of $p_z$ orbitals removed from each triangular sublattice. Single magnetic states close to $E_F$ can therefore be induced by removing $p_z$ orbitals from one of the sublattices[74-78]. The double occupation of such a magnetic state can be observed as the dual peaks with an energy splitting near $E_F$, owing to Coulomb repulsion U[71]. In our THPB-HOF, the $p_z$ orbitals can be removed by unbalanced molecule-substrate interaction in faulted/unfaulted halves, resembling to two sublattices of graphene. In this regard, the single and double DOS peaks, closed to $E_F$ observed on THPB-HOF, could indicate the appearance of single and double-occupied local magnetic states, respectively. The energy separation of ≈48 meV between double peaks in Fig. 4e, is comparable to the 63 meV for boron-doped[72] and 21 meV for hydrogen attached graphene[71]. The LDOS difference between "faulted halves" and "unfaulted halves" in differential conductance maps of THPB-HOF (Supplementary Note 7 and Fig. 9) indicates the interaction between the HOPG substrate and THPB monolayer. The LDOS of magnetic states is more prominent at the THPB molecule in "unfaulted half" and the edges of THPB molecule

in "faulted half" than elsewhere, manifesting the internal spatial distribution of spin states for molecules.

In conclusion, large-scale floating HOF/graphene heterostructure has been successfully synthesized via the self-lifting procedure on HOPG surface by UHV molecule deposition. The unique HOF structure and highly dispersive linear bands are charactered by STM and ARPES. A number of non-dispersive bands appear at deeper binding energies, where the two topmost sets of valence bands are attributed to the upper dual topological flat bands and the lower Dirac bands from THPB-HOF lattice. The narrow bandwidth and the reduced carrier velocity are the results of the large cell size and weak hopping via hydrogen bonds. The observed DOS peaks in tunneling spectroscopy manifest the emergence of local magnetic states induced by removing $p_z$ orbitals from $\pi$-conjugated lattice, similar to the cases of hydrogen or boron chemisorbed graphene.

## Methods
### Sample preparation and STM/STS characterization
The HOPG substrates were cleaved in UHV to obtain a clean surface before deposition of THPB molecules. THPB molecules were subsequently dosed onto HOPG single-crystal surface held at a temperature of ≈300 K at a base pressure better than $2 \times 10^{-10}$ Torr. The desired 2D HOF were obtained by maintaining the rate of evaporation of THPB molecules in the range of 0.06–0.07 monolayer (ML) per min and followed with the annealing of 1 h at room temperature.

STM experiments were carried out on an UHV commercial STM system (Unisoku) which can reach a low temperature of 400 mK by using a single-shot $^3$He cryostat. The base pressure was $2.0 \times 10^{-10}$ Torr. The THPB samples were deposited at the preparation chamber and then transferred in situ into STM chamber. All the STM measurements were performed at the temperature of 77 K. A commercial Pt−Ir tip was carefully prepared by e-beam heating. The STS and d$I$/d$V$ mapping

were obtained by the standard lock-in method by applying an additional small AC voltage with a frequency of 973.0 Hz. The d$I$/d$V$ spectra were collected by disrupting the feedback loop and sweeping the DC bias voltage. WSxM software was used for post processing of all STM data[79].

## ARPES measurement

The ARPES measurements were performed at the Vacuum Interconnected Nanotech Workstation of Suzhou Institute of Nano-Tech and Nano-Bionics (SINANO) with a ScientaOmicron DA30L analyzer and monochromatized He I$\alpha$ (h$\nu$ = 21.218 eV) light source. The samples were grown in situ and measured at $T$ = 77 K (unless otherwise specified) with a background vacuum better than $5 \times 10^{-11}$ mbar.

## In-situ Raman measurement

In-situ Raman measurements were performed by transferring the samples via the UHV tubes (base pressure better than $2 \times 10^{-10}$ mbar) in Nano-X at room temperature. And the Raman spectra were obtained under the back-scattering geometry by a confocal micro-Raman system (Horiba IHR550 system) with 532 nm excitation.

## DFT calculations

Our first-principles DFT calculations were performed with local-density-approximation (LDA)[80] using Vienna ab initio Simulation Package[81] code. An energy cutoff of 520 eV and a $4 \times 4 \times 1$ Monkhorst–Pack k-point grid were used[82]. The THPB-HOF monolayer contains 90 atoms per unit cell. For structural optimization calculation of the freestanding THPB-HOF monolayer, we fixed the lattice constants to experimental confirmed values ($a = b = 16.5$ Å) and only relaxed atomic positions. The structure was optimized until the atomic forces are smaller than 0.001 eVÅ$^{-1}$ and the change of total energy per cell is smaller than $10^{-5}$ eV. For calculations of THPB-HOF on top of graphene and four graphite layers, the structure was fully relaxed without any restriction. To account for the van der Waals interaction, the optB88-vdW method[83] was adopted. The energy and force convergence criteria for THPB-HOF/graphene (graphite) were set to $10^{-4}$ eV and 0.01 eVÅ$^{-1}$ respectively.

## Data availability

The data that support the findings of this study are available from the corresponding authors upon request. The optimized structure of the THPB-HOF monolayer on top of the eight-layered HOPG substrate (OSHOF/HOPG) Data sets. *figshare* https://figshare.com/s/4c8ad015173911c5fddc (2024)

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

## Acknowledgements
Sample growth and ARPES measurements were conducted at the Vacuum Interconnected Nanotech Workstation (Nano-X) in SINANO, Suzhou. STM work was conducted at School of Physics, Huazhong University of Science and Technology in Wuhan. The calculation was done on the CHPC at the University of Utah and US-DOE-NERSC. X.Z, M.H.P. and F.-S.L. acknowledge financial support from the National Natural Science Foundation of China (22372096, 11574095, 91745115, 22102129 and 12134008). X.Y.L. and F.L. acknowledge financial support from DOE-BES (No. DE-FG02-04ER46148). F.-S.L. also acknowledge financial support from the Suzhou Science and Technology Program (Grant No. SJC2021009) and the Youth Innovation Promotion Association of Chinese Academy of Sciences (2017370). J.Z.G. acknowledge the support program for young top-notch talents in Shaanxi Province (1511000066) and the National Development and Reform Commission (NDRC) of the People's Republic of China special subsidy for the preliminary work of the Western Development. X.L. acknowledges also the National Science Foundation (No. 2326228).

## Author contributions
X. Zhang, X. Y. Li, Z. W. Cheng, A. X. Chen and P. D. Wang contributed equally to this work. F.-S. Li, F. Liu and M. H. Pan conceived and designed the experiment. X. Zhang, Q. Bian and X. Y. Wang grew the samples. Z. W. Cheng, S. J. Li, B. Qi and J. Gao performed the STM experiments. A. X. Chen, X. X. Lei, P. D. Wang and F.-S. Li performed the ARPES measurements. X. Y. Li carried out the calculations. B. K. Yuan performed the measurement on $MoS_2$ substrate. M. H. Pan and F. Liu wrote the manuscript with inputs from all other authors.

## Competing interests
The authors declare no competing interests.
