## [Peer Review File · Nature Communications]

REVIEWER COMMENTS

Reviewer #1 (Remarks to the Author):

The authors provide a method for constructing 2D large-scale organic framework/graphene heterostructures with observed Dirac and Flat bands in this paper. The researchers create a well-ordered THPB-HOF on a HOPG substrate and notice that the topmost layer of HOPG is self-lifted to generate floating HOF/graphene heterostructures due to the strong interlayer coupling force between graphene and HOF. They characterize the distinctive structure and dispersive linear bands using STM and ARPES, and they aid in DFT calculations for further result analysis and simulation. Overall, the findings are promising. However, in my opinion, the authors need to provide a more extensive explanation for how their method represents a technological progress in order for it to be published in Nature Communications, and several questions need to be further solved. Detailed comments are provided below.

1. As the authors demonstrate in line 57 of page 3, it is usual to generate high quality organic-inorganic heterostructures, such as highly-ordered organic layers on graphene, MoSe₂, and WS₂, using the vapour phase growth approach. Furthermore, on Au and Ag substrates, the self-assembled superstructure of THPB molecules is formed. In terms of the preparation process, it appears that the innovation is insufficient; the uniqueness of the method is not adequately expressed in this work.
2. It is better to provide the details of graphene structure in supplementary data for the black region of STM image in Figure 1b and c by changing the colour difference, and I'm intrigued how the sharp boundary is obtained. Furthermore, under the same test settings, the surface structure appears to be different in Figure 1b and 1c. What could be the reason for the structural differences? Various HOF thicknesses, or the possibility of Moiré patterns between HOF/Graphene and HOPG.
3. In Figure 1d, the author calculated the HOF on four layers of HOPG; why four layers? In Figure 1b-c, the STM measurement results show a double and monolayer layer. What are the outcomes for the 1 to 3 layer HOPG calculation?
4. Is it possible to peel THPB/graphene onto a substrate such as SiO₂/Si for Raman characterization? What I don't understand is why 2D peaks vanish when the HOPG comprises of many carbon layers. In addition, the measured intrinsic HOPG should be provided for comparison. The difference between the Raman peaks on graphene (3268 and 3398 cm⁻¹) and on Au substrate (3117 and 3226 cm⁻¹) in Extended data Fig. 2a and 2b is so enormous that it is difficult not to suspect that peak may have different origins. There are also other Raman peaks in Extended data Fig. 2a that need to be explained.
5. The reason for bands δ_1 to δ_5 in ARPES in Figure 3f is obtained from hypothesis or has been indicated in other works in line 204-205, page 8.

6. It is suggested that the author combine the Extended Data with the Supplementary figure to create the new Supplementary figure. Some of the figures, such as Fig 1d and Extended Data Fig 1b and d, are repeated in the text and supplementary data. More importantly, several supplemental data, such as supplementary Figs. S6 and S7, are not mentioned or analysed in the text in the correct order.

Reviewer #2 (Remarks to the Author):

2D organic/inorganic heterostructure is an important topic as it may provide complementary properties compared to 2D inorganic hetero-/homo-structures. Zhang et al. have reported a facile method aiming to achieve 2D organic-inorganic heterojunctions. Typically, 1,3,5-tris(4-hydroxyphenyl)benzene (THPB) was directly deposited on highly-oriented-pyrolytic-graphite (HOPG), and then was in-situ annealed, resulting in a strong interlayer coupling between THPB and the topmost layer of HOPG. Density-of-state (DOS) peaks near EF and Dirac bands of THPB/HOPG heterojunction were observed and discussed. This work makes a decent progress in feasibly realizing organic-inorganic heterojunctions. I would like to recommend it to be published in Nature Communication, after revisions to address the following concerns.

1. Considering 'Large-scale heterostructure' is announced in the title, optical microscope photos are suggested to add to help highlight a large-scale organic-inorganic heterojunction realized by this method. Recent representative works of organic-inorganic heterojunctions with strong interlayer coupling are suggested to be added in Introduction part, for example a recent review (SmartMat 2023, 4 (2), e1156). Meanwhile, a brief discuss about uniqueness of organic/inorganic heterostructures is needed.
2. To further investigate properties of such heterojunction at a 2D limit, exfoliating is important. Are there any results of an exfoliated heterojunction from HOPG? Please discuss.
3. Why are band structures above E_f , shown in Fig.3e, not observed in Fig.3c? Please discuss.
4. Discussion of Fig.4 in 'DFT calculations of topological flat bands and Dirac bands of THPB-HOF' part is partially repeating with that in 'ARPES observation of Dirac and narrow bands in THPB-HOF/HOPG' part. Fig.4 is suggested to merge into Fig.3 to help explain ARPES results. Please consider make them brief. Please correct the format problems in the reference.
5. As authors mentioned that STS is related to local DOS, how to understand that cells have only one DOS peak near E_f , while others 'occasionally' have two DOS peaks?
6. Attempts on other vdW materials are suggested to supply or discussed to show a universality of this method.

With the above improvements, the work is expected to generate a high impact in the communities of both 2D materials and condense matter physics.

Reviewer #4 (Remarks to the Author):

The manuscript presents a novel approach to build 2D organic-inorganic heterostructures, aiming to address shortcomings seen in traditional methods. Employing a bottom-up technique in an ultrahigh vacuum (UHV) setting, the study strives to develop extensive heterostructures with promising features like immaculate interfaces and remarkably crystalline sheets. Although the findings on the 1,3,5-tris(4-hydroxyphenyl)benzene (THPB)/graphene systems appear coherent, I have a few queries/comments that need clarification before considering recommending the manuscript for publication.

#1 - The method for bottom-up fabrication of 2D large-scale organic-inorganic structures appears quite effective for 1,3,5-tris(4-hydroxyphenyl)benzene (THPB). I'm curious about its suitability for other HOFs, and possibly even for MOFs and COFs. Are there indications from the authors that this approach is feasible for these materials? Additionally, could they provide evidence and outline the essential conditions necessary to prepare another hybrid system using a comparable approach?

2 - The authors frequently refer to the synthesis approach producing a "large-scale" heterostructure." It would be beneficial if they could offer more precise specifications and quantitative details regarding this large-scale structure. It could be insightful to compare this scale with those achieved through other state-of-the-art approaches for clarity and a comprehensive assessment.

3 - Given that the manuscript primarily centres on the methodology for crafting 2D organic-inorganic heterostructures, could the authors provide more clarity on the rationale behind selecting the 1,3,5-tris(4-hydroxyphenyl)benzene (THPB) system for their study?

4 - The authors achieved the successful synthesis of the HOF/graphene heterostructure through a self-lifting technique on the HOPG surface, employing UHV molecule deposition. Notably, STM and ARPES characterization's unveil the HOF structure and highly dispersive linear bands. To what extent do these findings rely on the specific synthesis approach used? It would be insightful if the authors could offer a comparison with HOF/graphene grown via alternative methods for a more comprehensive assessment.

Responds to Referee #1

Reviewer #1 (Remarks to the Author):

The authors provide a method for constructing 2D large-scale organic framework/graphene heterostructures with observed Dirac and Flat bands in this paper. The researchers create a well-ordered THPB-HOF on a HOPG substrate and notice that the topmost layer of HOPG is self-lifted to generate floating HOF/graphene heterostructures due to the strong interlayer coupling force between graphene and HOF. They characterize the distinctive structure and dispersive linear bands using STM and ARPES, and they aid in DFT calculations for further result analysis and simulation. Overall, the findings are promising. However, in my opinion, the authors need to provide a more extensive explanation for how their method represents a technological progress in order for it to be published in Nature Communications, and several questions need to be further solved. Detailed comments are provided below.

Authors' reply: We thank the referee for the overall positive evaluation of our work. Below, we address the comments raised by this referee point-to-point and show the revisions made in the manuscript accordingly.

In order to demonstrated the universality of our method, we prepared the THPB monolayer on MoS_2 substrate. Indeed, we observed the bandgap transition of MoS_2 from bulk to monlayer via similar self-lifting effect induced by strong interlayer coupling with THPB monolayer.

1. As the authors demonstrate in line 57 of page 3, it is usual to generate high quality organic-inorganic heterostructures, such as highly-ordered organic layers on graphene, MoSe_2 , and WS_2 , using the vapour phase growth approach. Furthermore, on Au and Ag substrates, the self-assembled superstructure of THPB molecules is formed. In terms of the preparation process, it appears that the innovation is insufficient; the uniqueness of the method is not adequately expressed in this work.

Authors' reply: We thank the referee for this constructive comment. Our work brings the novelty and advancement at least in the following aspects:

1. Presenting a simple but effective method to build 2D "large-scale" organic-inorganic heterostructure. Please note most previous works achieve high quality or highly-order samples but usually with much smaller sample size.
2. Revealing a self-lifting effect on the topmost layer of HOPG via strong interlayer coupling induced by HOF monolayer, to form effectively a floating HOF/graphene heterostructure. This phenomenon is new, to the best of our knowledge.
3. Directly observing the coexistence of distinct Dirac bands with high carrier mobility from graphene and flattened bands of HOF is new, which is enabled by a new growth

mechanism with enhanced organic-inorganic interaction to self-lifted heterostructure as mentioned above in point 2.

4. In addition, we have now added new results to demonstrate the generality of our method by growing HOF on a different MoS_2 substrate. Again, we observed the bandgap transition of MoS_2 from bulk to monolayer via similar self-lifting effect induced by strong interlayer coupling with THPB monolayer.

In page 3 line 78, to further convey the novelty of our work, we added “In short, we present a simple effective method to make freestanding 2D large-scale organic-inorganic heterostructure by a self-lifting effect, as confirmed by directly observing the coexistence of distinct Dirac bands with high carrier mobility from graphene and flattened bands of HOF due to enhanced organic-inorganic interaction. We also further demonstrated the generality of our method by growth on MoS_2 substrate, where we observed the bandgap transition of MoS_2 from bulk to monolayer via again the self-lifting effect induced by THPB-HOF monolayer.”.

2. It is better to provide the details of graphene structure in supplementary data for the black region of STM image in Figure 1b and c by changing the colour difference, and I'm intrigued how the sharp boundary is obtained. Furthermore, under the same test settings, the surface structure appears to be different in Figure 1b and 1c. What could be the reason for the structural differences? Various HOF thicknesses, or the possibility of Moiré patterns between HOF/Graphene and HOPG.

Authors' reply: Following the reviewer's suggestion, we replotted Fig. 1b and 1c by changing the color difference. To visualize the details of underneath graphene structure, we zoom into the yellow square region in **Fig. R1b**. As one can see, the atomic-resolution image of graphene lattice in **Fig. R1c** shows a perfect triangular lattice, indicating it comes from graphite, according to the triangular/honeycomb atomic lattice transition phenomenon. [Wong, H. S.; Durkan, C.; Chandrasekhar, N. Tailoring the Local Interaction between Graphene Layers in Graphite at the Atomic Scale and Above Using Scanning Tunneling Microscopy. *ACS Nano* 2009, **3** (11), 3455–3462.]. The $\langle 1101 \rangle$ direction of graphite (yellow dashed lines in **Fig. R1c**) is perfectly aligned with HOF lattice direction (red dashed diamond in **Fig. R1d** and **1e**), indicating the graphite substrate is uplifted without covering the top HOF structure. These results are consistent with the height of 6.48 Å in **Fig 1d** (corresponding to the thickness of top HOF layer and top graphene layer) as well as the height of 2.65 Å in **Fig. 1d** (corresponding to the thickness of only top HOF layer).

We also zoom into the HOF regions (marked with red squares) in **Fig. R1a** and **R1b**, which are shown in **Fig. R1d** and **R1e**, respectively. One sees that both HOF have the same structures; the slight difference in the images are likely due to a lower resolution of **R1d**.

Fig. R1 a, replotted Fig. 1b b, replotted Fig. 1c; c, the atomic resolution image of graphene lattice zoomed into the yellow square region in 1b. d and e, zoom-in images of HOF regions (marked with red squares) in Fig. R1a and R1b.

3. In Figure 1d, the author calculated the HOF on four layers of HOPG; why four layers? In Figure 1b-c, the STM measurement results show a double and monolayer layer. What are the outcomes for the 1 to 3 layer HOPG calculation?

Authors' reply: We used four layers just to save computational time. The results of this particular case clearly showed the enhanced interaction between the THPB-HOF monolayer and the first layer of HOPG as well as the weakened interaction between the first layer and the remaining layers of HOPG, demonstrating the self-lifting effect of THPB-HOF monolayer to exfoliate graphene from HOPG. Although we used this particular case as a demonstration, we stress that the self-lifting effect remains the same for different thickness of HOPG. To prove that, during the revision, we calculate the THPB-HOF monolayer on the thicker eight-layered HOPG. The calculated results are plotted in Fig. R2. One can see that after structural optimization, the interlayer distance between the THPB-HOF monolayer and the first layer of HOPG is reduced while the interlayer distance between the first layer and the remaining layers of HOPG is enlarged, qualitatively consistent with the results of Fig. 1d. The numerical discrepancy could be attributed to the different settings for structural optimization calculation. Here for the eight-layered HOPG calculation, we fix the atomic positions of the bottom two layers and relax all other atoms to mimicking an infinite thick HOPG substrate meanwhile reducing the computational cost. In **Figure 1b-c**, actually STM measurement shows a layer of THPB-HOF plus a layer of graphene in 1b and just a THPB-HOF layer in 1c, respectively. The calculation of 1 to 3 HOPG layer would be unphysical or unrealistic, so we didn't do it.

To make this point clearer, we have replaced the Extended Data Fig. 1 with Fig.

R2 and added the following sentence to the revised manuscript:

In page 4 Line 111, we add “Here the four-layered HOPG is used as a manifestation, but we emphasize that the self-lifting effect of THPB-HOF monolayer and the resultant exfoliation of graphene are robust regardless of the thickness of HOPG (see Extended Data Fig. 1).”

Fig. R2 Structural relaxation of the THPB-HOF monolayer on top of the eight-layered HOPG substrate. **a-b**, top and side view of the initial structure. **c-d**, same as **a-b** but for the final optimized structure. Here the atomic positions of the bottom two layers of HOPG (highlighted by the pink shadow) are fixed and all other atomic positions are relaxed during the structural optimization calculation, to mimicking an infinite thick HOPG substrate meanwhile reducing the computational cost.

4. Is it possible to peel THPB/graphene onto a substrate such as SiO₂/Si for Raman characterization? What I don't understand is why 2D peaks vanish when the HOPG comprises of many carbon layers. In addition, the measured intrinsic HOPG should be provided for comparison. The difference between the Raman peaks on graphene (3268 and 3398 cm⁻¹) and on Au substrate (3117 and 3226 cm⁻¹) in Extended data Fig. 2a and 2b is so enormous that it is difficult not to suspect that peak may have different origins. There are also other Raman peaks in Extended data Fig. 2a that need to be explained.

Authors' reply: We thank the referee for this interesting comment. To peel THPB/Graphene onto SiO₂/Si substrate will be challenging technically, due to the contamination induced by the exposure to atmosphere will attach to both THPB layer and Graphene layer. The disappearance of the 2D peak of graphene (~2689 cm⁻¹) may be caused by the strong interlayer interaction between THPB-HOF and graphene. As we know, the Raman spectroscopy is only sensitive to the topmost layers. The enormous difference between the Raman peaks on graphene (3268 and 3398 cm⁻¹)

and on Au substrate (3117 and 3226 cm^{-1}) in Extended data Fig. 2a and 2b indicates the strength of hydrogen bonds in HOF on graphite is much weaker than that on Au(111), since they are characteristic peaks of H bonds. In our previous calculations, the peak at 3474 cm^{-1} is identified from the H-O vibration, whose position is indicative of the H-bond strength [Y. Ling, *et al.*, *Sci. Rep.* **6**, 31981 (2016)]. To confirm this, we also calculated the Raman spectrum of the THPB-HOF under a 2% biaxial tensile strain (see Fig. R2). The position of the H-O peak of the strained HOF changes to 3584 cm^{-1} , indicating a blue shift of 110 cm^{-1} due to the weakened H-bonds, while the other peaks remain almost intact.

Fig. S11 Calculated Raman spectra of the pristine THPB-HOF on Au(111) (a) and the 2% biaxial tensile strained THPB-HOF (b). (reproduced from **Supplemental Material of Ref. 37**)

For comparison, as the referee suggested, we also measured Raman spectroscopy on intrinsic HOPG substrate. As we see in **Fig. R2a**, Both G ($\sim 1588 \text{ cm}^{-1}$) and 2D ($\sim 2689 \text{ cm}^{-1}$) peaks shows up clearly. AT the same time, we also measured Raman spectroscopy on THPB-HOF prepared on bilayer Graphene(BLG) on SiC (**Fig. R2b**) for comparison. the mode of 1637 cm^{-1} and $\sim 2700 \text{ cm}^{-1}$ can be assigned to the G peak of graphene ($\sim 1588 \text{ cm}^{-1}$) with an upshift of 49 cm^{-1} and the 2D peak of graphene (2689 cm^{-1}) with an upshift of 11 cm^{-1} . Furthermore, Two characteristic Raman peaks (3108 and 3248 cm^{-1}), similar to the modes (3117 and 3226 cm^{-1}) on Au(111), can be assigned to the H-O vibration.

Fig. R2. a, Raman spectroscopy measured on intrinsic HOPG and THPB on bilayer graphene on SiC(**b**).

5. The reason for bands $\delta 1$ to $\delta 5$ in ARPES in Figure 3f is obtained from hypothesis or has been indicated in other works in line 204-205, page 8.

Authors' reply: The reason for labeling bands $\delta 1$ to $\delta 5$ in ARPES in Figure 3f, actually is motivated from the calculation of flat bands of freestanding THPB-HOF in Fig. 4b, not from hypothesis in other work. Now we add the label of bands $\delta 1$ to $\delta 5$ in Fig. 3f.

6. It is suggested that the author combine the Extended Data with the Supplementary figure to create the new Supplementary figure. Some of the figures, such as Fig 1d and Extended Data Fig 1b and d, are repeated in the text and supplementary data. More importantly, several supplemental data, such as supplementary Figs. S6 and S7, are not mentioned or analysed in the text in the correct order.

Authors' reply: We thank the referee for this constructive comment. In the revised version, we replace the previous repeated Extended Data Fig 1 with the new calculated results of THPB-HOF monolayer on top of the eight-layered HOPG (see our response to the reviewer's comment #3).

We also rearrange the order of Figs. S5, S6 and S7 in SI.

In line 160, we add “Similar contrast of THPB-HOF can be clearly visualized in STM images with different biases in Supplementary Fig. S5. More interestingly, we also observed two different kinds of chirality for H-bond hollow-ring in STM imaging of THPB-HOF, as shown in Fig. S6a and S6b.”

Responds to Referee #2

Reviewer #2 (Remarks to the Author):

2D organic/inorganic heterostructure is an important topic as it may provide complementary properties compared to 2D inorganic hetero-/homo-structures. Zhang et al. have reported a facile method aiming to achieve 2D organic-inorganic heterojunctions. Typically, 1,3,5-tris(4-hydroxyphenyl)benzene (THPB) was directly deposited on highly-oriented-pyrolytic-graphite (HOPG), and then was in-situ annealed, resulting in a strong interlayer coupling between THPB and the topmost layer of HOPG. Density-of-state (DOS) peaks near EF and Dirac bands of THPB/HOPG heterojunction were observed and discussed. This work makes a decent progress in feasibly realizing organic-inorganic heterojunctions. I would like to recommend it to be published in Nature Communication, after revisions to address the following concerns.

Authors' reply: We thank the referee for the positive evaluation and recommendation of our work. Below, we response and address the comments raised by this referee point-to-point and revise the manuscript accordingly.

1. Considering 'Large-scale heterostructure' is announced in the title, optical microscope photos are suggested to add to help highlight a large-scale organic-inorganic heterojunction realized by this method. Recent representative works of organic-inorganic heterojunctions with strong interlayer coupling are suggested to be added in Introduction part, for example a recent review (SmartMat 2023, 4 (2), e1156). Meanwhile, a brief discuss about uniqueness of organic/inorganic heterostructures is needed.

Authors' reply: We thank the referee for this constructive comment.

Now we add the optical microscope photos of sample in Fig. R3 as shown below. As one sees from the image, the flat surface area of HOPG marked by yellow dashed lines is about 2x3 mm². Within such area, the uniform THPB-HOF could form. Unfortunately, the optical microscope cannot image THPB-HOF structure.

Fig. R3 the optical image of HOPG sample. HOPG surface is obtained via vacuum-cleavage.

We also add this into SI as **Fig. S13**.

And the review of (SmartMat 2023, 4 (2), e1156) is also added

16. Khan, J, Ahmad, RTM, Tan, J, Zhang, R, Khan, U, Liu, B. Recent advances in 2D organic-inorganic heterostructures for electronics and optoelectronics. *SmartMat.* 2023; **4**, e1156.

In page 2 line 48, a brief discuss about uniqueness of organic/inorganic heterostructures is also added: “the organic-inorganic heterostructures may take the advantages of both organic and inorganic materials, for example to make it flexible like an organic film and at the same time wearable like an inorganic film. The combination of their different properties and functionalities may broaden the device's capabilities and endow special applications that neither one can achieve alone.”.

2. To further investigate properties of such heterojunction at a 2D limit, exfoliating is important. Are there any results of an exfoliated heterojunction from HOPG? Please discuss.

Authors' reply: To peel THPB/Graphene onto SiO₂/Si substrate will be very challenging technically, due to the contamination induced by the exposure to atmosphere will attach to both THPB layer and Graphene layer. On the other hand, mechanical exfoliation by Scotch-tape is not appropriate due to the graphene layer is fully covered by THPB-HOF layer. Other exfoliation method, such as the inserting of an organic intercalation layer will also introduce the contamination. We are not aware of any exfoliated heterojunction from HOPG. In our set up of UHV condition, maybe some in situ transfer techniques could be developed in the future.

3. Why are band structures above E_f, shown in Fig.3e, not observed in Fig.3c? Please

discuss.

Authors' reply: Angle-resolved photoelectron spectroscopy (ARPES) can only measure the bands below E_F .

4. Discussion of Fig.4 in 'DFT calculations of topological flat bands and Dirac bands of THPB-HOF' part is partially repeating with that in 'ARPES observation of Dirac and narrow bands in THPB-HOF/HOPG' part. Fig.4 is suggested to merge into Fig.3 to help explain ARPES results. Please consider make them brief. Please correct the format problems in the reference.

Authors' reply: We thank the referee for the suggestion. Now we merged **Fig. 4** into **Fig.3**, and the revised Fig. 3 is shown below.

Fig. R4. The revised Fig.3

5. As authors mentioned that STS is related to local DOS, how to understand that cells have only one DOS peak near E_f , while others 'occasionally' have two DOS peaks?

Authors' reply: We thank the referee for bringing up this interesting question, which deserves further study. Our current thinking is mentioned in the main text that the double occupation of a magnetic state can be observed as the dual DOS peaks with an energy splitting near E_F , owing to Coulomb repulsion U , similar to Fig.1C in [Science 352, 437–439 (2016)].

7. Attempts on other vdW materials are suggested to supply or discussed to show a

universality of this method.

Authors' reply: We thank the referee for this constructive suggestion. In order to show the generality of this method, we tried this method on MoS₂ substrate. MoS₂ is a promising vdW material. We deposited THPB molecule on *in-situ*-cleaved MoS₂ surface. **Fig. R3** and **R4** show the morphology of THPB molecules on MoS₂ surface measured by STM. The thickness of monolayer THPB framework is measured about 2.94 Å, similar to 2.65-2.71Å on HOPG substrate. High resolution STM image in **Fig. R4b** shows the similar HOF structure with unit cell size about 2.67 nm, larger than the size of unit cell of THPB-frame on HOPG (16.5 Å). Most importantly, we measured differential conductance spectroscopy dI/dV on both clean MoS₂ surface and THPB molecule island (**Fig. R4c-d**). the bandgap of MoS₂ clean surface is measured about 1.46 eV, smaller than the bandgap (1.78 eV) measured on THPB-framework, indicating the self-lifting effect of THPB-framework also acts on the MoS₂ layer. The bulk MoS₂ material is reported to have an indirect bandgap of 1.2 eV, whereas two-dimensional (2D) single-layer MoS₂ nanosheets have a direct bandgap of 1.8 eV [36,37]. Our results are consistent with the reported bandgap transition of MoS₂ from bulk to monolayer. These new results evidently support the generality of our method.

In page 5 line 124, we add "We also tried deposit THPB molecule on *in-situ*-cleaved MoS₂ surface. Supplementary **Fig. S11** and **S12** show the morphology of THPB molecules on MoS₂ surface measured by STM. The thickness of monolayer THPB framework is measured about 2.94 Å, similar to 2.65-2.71Å on HOPG substrate. High resolution STM image in Supplementary **Fig. S12b** shows the similar HOF structure with unit cell size about 2.67 nm, larger than the size of unit cell of THPB-framework on HOPG (16.5 Å). Most notably, we measured differential conductance spectroscopy (dI/dV) on both clean MoS₂ surface and THPB molecule island (Supplementary **Fig. S12c-d**). the bandgap of MoS₂ clean surface is measured about 1.46 eV, slightly smaller than the bandgap (1.78 eV) measured on THPB-framework, indicating the self-lifting effect of THPB-framework acts also on the MoS₂ layer. The bulk MoS₂ material is reported to have an indirect bandgap of 1.2 eV, whereas two-dimensional (2D) single-layer MoS₂ nanosheets have a direct bandgap of 1.8 eV [36,37]. Our results are consistent with the reported bandgap transition of MoS₂ from bulk to monolayer. These results evidently support the generality of our method."

In SI, we add this as Fig. **S11** and **S12**.

Fig. R3 Large-scale STM image of THPB molecular island formed on MoS₂ clean surface. **a**, THPB island formed on vacuum-cleaved MoS₂ surface. The image are 100×100 nm² with the setting parameters of V_B : -5V and I_T : 5pA. **b**, the height profile measured along the red line in panel **a**, which shows the thickness of monolayer THPB is about 2.94 Å.

Fig. R4 High resolution STM images and dI/dV tunneling spectroscopic measurements on THPB on MoS₂. **a**, large scale image show THPB island formed on vacuum-cleaved MoS₂ surface. The image are 200×200 nm² with the setting parameters of V_B:-5V and I_T:5pA. **b**, high-resolution image of THPB framework on MoS₂. The image are 10×10 nm² with the setting parameters of V_B:-5V and I_T:5pA. The image shows similar structure as the THPB-HOF on HOPG, the size of unit cell is about 2.67 nm, larger than the size of unit cell of THPB-frame on HOPG (16.5 Å). **c-d**, dI/dV spectrum measured on clean MoS₂ surface (cyan) and THPB framework (blue), respectively, **c**, data plotted as linear coordination and **d**, data plotted as log coordination. The location of STS measurements are marked as points with corresponding color in panel **a**. Actually, the bandgap of MoS₂ clean surface is measured about 1.46 eV, smaller than the bandgap (1.78 eV) measured on THPB-framework, indicating the self-lifting effect of THPB-framework applying on the MoS₂ layer. The bulk MoS₂ material is reported to have an indirect bandgap of 1.2 eV, whereas two-dimensional (2D) single-layer MoS₂ nanosheets have a direct bandgap of 1.8 eV [36,37]. Our results are consistence with the reported bandgap transition of MoS₂ from bulk to monolayer. This result provides the evidence of the universality of our method.

Reference:

36. A. Splendiani, L. Sun, Y. Zhang, T. Li, J. Kim, C.-Y. Chim, G. Galli, F. Wang. Emerging photoluminescence in monolayer MoS₂. *Nano Lett.*, **10** (2010), 1271-1275
37. K.F. Mak, C. Lee, J. Hone, J. Shan, T.F. Heinz. Atomically thin MoS₂: a new direct-gap semiconductor. *Phys. Rev. Lett.*, **105** (2010), 136805

With the above improvements, the work is expected to generate a high impact in the communities of both 2D materials and condense matter physics.

Reviewer #4 (Remarks to the Author):

The manuscript presents a novel approach to build 2D organic-inorganic heterostructures, aiming to address shortcomings seen in traditional methods. Employing a bottom-up technique in an ultrahigh vacuum (UHV) setting, the study strives to develop extensive heterostructures with promising features like immaculate interfaces and remarkably crystalline sheets. Although the findings on the 1,3,5-tris(4-hydroxyphenyl)benzene (THPB)/graphene systems appear coherent, I have a few queries/comments that need clarification before considering recommending the manuscript for publication.

Authors' reply: We thank the referee for the positive evaluation and recommendation of our work. Below, we response and address the comments raised by this referee point-to-point and revise the manuscript accordingly.

#1 - The method for bottom-up fabrication of 2D large-scale organic-inorganic

structures appears quite effective for 1,3,5-tris(4-hydroxyphenyl)benzene (THPB). I'm curious about its suitability for other HOFs, and possibly even for MOFs and COFs. Are there indications from the authors that this approach is feasible for these materials? Additionally, could they provide evidence and outline the essential conditions necessary to prepare another hybrid system using a comparable approach?

Authors' reply: We thank the referee for this constructive suggestion, which is also suggested by the second referee. In order to show the generality of this method, we tried also this method on MoS₂ substrate, as well as the epitaxially-grown graphene on SiC. Please see also our reply to Question 7 of referee #2 and Question 4 of this referee. We will leave the possibilities of MOFs and COFs for possible future works.

2 - The authors frequently refer to the synthesis approach producing a "large-scale" heterostructure." It would be beneficial if they could offer more precise specifications and quantitative details regarding this large-scale structure. It could be insightful to compare this scale with those achieved through other state-of-the-art approaches for clarity and a comprehensive assessment.

Authors' reply: We thank the referee for this constructive suggestion. From large scale STM showing in Fig. S4a, the HOF is at least uniform and mono-orientated in the range of 125x125 nm². And in our reply to question #1 of referee #2, We also provide the optical microscope photos of HOPG sample. As one can see from the image, the flat surface area of HOPG marked by yellow dashed lines is about 2x3 mm². Within such area, in principle, the uniform THPB-HOF could grow.

3 - Given that the manuscript primarily centres on the methodology for crafting 2D organic-inorganic heterostructures, could the authors provide more clarity on the rationale behind selecting the 1,3,5-tris(4-hydroxyphenyl)benzene (THPB) system for their study?

Authors' reply: We thank the referee for this constructive comment. The choice is made mostly based on our previous experience on growing THPB molecular film. First, we found THPB is able to form uniform HOF structure via strong hydrogen bond connecting each other. Second, strong interactions are expected between HOF layer and substrate. THPB molecule has numerous benzene rings. When stack it on to graphite, a π - π stacking will be formed. An early explanation of π - π interactions comes from the Hunter-Sanders electrostatic model, which proposes that two benzene rings in a face-to-face sandwich configuration combine together mainly due to the repulsion between their π electrons and the attraction between the positively charged benzene rings and their π electrons. [C. A. Hunter and J. K. M. Sanders, *J. Am. Chem. Soc.*, 1990, **112**, 5525—5534; F. Cozzi, M. Cinquini, R. Annunziata, T. Dwyer and J. S. Siegel, *J. Am. Chem. Soc.*, 1992, **114**, 5729—5733]. Therefore, similarly the π - π interactions will likely to form strong interactions between HOF layer and substrate.

4 - The authors achieved the successful synthesis of the HOF/graphene heterostructure through a self-lifting technique on the HOPG surface, employing UHV

molecule deposition. Notably, STM and ARPES characterization's unveil the HOF structure and highly dispersive linear bands. To what extent do these findings rely on the specific synthesis approach used? It would be insightful if the authors could offer a comparison with HOF/graphene grown via alternative methods for a more comprehensive assessment.

Authors' reply: We thank the referee for this constructive suggestion. Now we also tried this HOF on the epitaxially-grown graphene on SiC. In **Extended Data Fig. 3**, we show ARPES observation of the THPB/BLG-SiC bands. Note, SiC surface provide high-quality bilayer graphene layer. However, by comparing the ARPES data from THPB/Graphite and THPB/BLG-SiC, we can find the linear bands of Graphene layer and FBs of THPB-HOF of THPB/graphite are even better resolved than those of THPB/BLG-SiC, which indicate the heterostructure interface created by our method is better than the method of directly deposit the molecule on epitaxial-grown graphene.

REVIEWERS' COMMENTS

Reviewer #1 (Remarks to the Author):

The revised version has already corrected all my comments for the first edition, and it seems to be more suitable for the publication in Nature Communications.

Reviewer #2 (Remarks to the Author):

I have carefully examined the revised version and found that all the previous points have been well addressed. Therefore, I support its publication in the current form. The work is expected to generate a high impact in the communities because of the unique growth method and the properties of the obtained heterostructures. Congratulations on the nice work.

Reviewer #4 (Remarks to the Author):

Dear Editor,

I carefully read the author's response and recognized their efforts in answering the questions diligently. After evaluation, I consider the manuscript suitable for publishing in Nature Communications.

Best regards,